# Fine Root Length of Maize Decreases in Response to Elevated CO₂ Levels in Soil

**Yaojie Han [1,2], Xueyan Zhang [1,3,*] and Xin Ma [2]**

[1]   Institute of Geographic Sciences and Natural Resources Research, Chinese Academy of Sciences, Beijing 100101, China; hanyaojie@aliyun.com
[2]   Institute of Environment and Sustainable Development in Agriculture, Chinese Academy of Agricultural Sciences, Beijing 100081, China; maxin02@caas.cn
[3]   Center for Chinese Agricultural Policy, Chinese Academy of Sciences, Beijing 100101, China
[*]   Correspondence: xyzhang@igsnrr.ac.cn; Tel.: +86-010-64888673



**Featured Application: The effect of CO₂ leaks from CCS on plant root systems was investigated; morphological characteristics of maize roots degraded by 19.16–44.73%; fewer fine roots in a root system was the main response to CO₂ leakage; and empirical evidence is essential to explaining plant responses to CO₂ leakage.**

**Abstract:** To assess the environmental risks of carbon capture and storage (CCS) due to underground CO₂ leakage, many studies have examined the impact on plant growth; however, the effect of leaked CO₂ on root morphology remains poorly understood. This study simulated the effects of CO₂ leakage from CCS on maize (*Zea mays L.*) root systems through pot experiments—one control treatment (no added CO₂) and two elevated soil CO₂ treatments (1000 g m⁻² d⁻¹ and 2000 g m⁻² d⁻¹). Compared with the control, root length, root surface area, and root volume were reduced by 44.73%, 34.14%, and 19.16%, respectively, in response to CO₂ treatments with a flux of 2000 g m⁻² d⁻¹. Meanwhile, the fine root length in CO₂ treatments with a flux of 1000 g m⁻² d⁻¹ and 2000 g m⁻² d⁻¹ were reduced by 29.44% and 45.88%, respectively, whereas no obvious difference in regard to coarse roots was found. Understanding changes in plant root morphology in this experiment, especially the decrease in the fine root length, are essential for explaining plant responses to CO₂ leakage from CCS.

**Keywords:** diameter class length; elevated CO₂; maize; root length; root diameter

## 1. Introduction

Carbon capture and storage (CCS) can capture up to 90% of the carbon dioxide (CO₂) emissions generated by fossil fuels in power generation and industrial processes, thereby reducing the amount of CO₂ entering the atmosphere [1]. Moreover, there are seventeen large-scale facilities that are already successfully in operation around the world (with four more scheduled to come online in the near future), which are currently capable of capturing more than 30 million tons of CO₂ per annum [2]. However, an unavoidable issue associated with CCS is the leakage of CO₂ from geological storage sites into the soil and atmosphere [3,4].

Many tests of CO₂ injection into soil have been carried out to assess the potential impacts on agro-ecological systems and plant growth [5–7]. Currently, injected CO₂ has been proven to reduce the soil O₂ concentration and decrease soil pH, which have negative impacts on plant growth [8–11]. For example, this was demonstrated at the ASGARD (Artificial Soil Gassing and Response Detection) facility site, where injected CO₂ significantly limited productivity in both grass and clover, visible stress symptoms were observed in plants exposed to high soil CO₂ concentrations, and the spectral

reflectance also changed [12,13]. This has also been shown in Korea, where a medium $CO_2$ treatment (i.e., with a soil $CO_2$ concentration of 12%) retarded plant germination, and no seeds germinated with a high $CO_2$ concentration (29.4%) [14]. Many assessments of the effects on plants due to $CO_2$ leakage have mostly focused on plant responses. A previous study demonstrated that, in response to a leakage treatment with a $CO_2$ flux of 2000 g m$^{-2}$ d$^{-1}$, the height of maize and alfalfa decreased by 35.71% and 52.25%, respectively, compared with the control, while similar responses to leaked $CO_2$ in terms of the number of visible leaves, unfolded leaves, and leaf area were also found [15]. Moreover, Sharma et al. found that the chlorophyll content (i.e., a unitless measure) of dandelions decreased significantly from 9.49 to 1.91 (i.e., by 80%) in response to the intentional release of 0.15 ton day$^{-1}$ $CO_2$ through a horizontal injection well that was 100 m long and 2.0–2.3 m deep [16]. Furthermore, elevated soil $CO_2$ lowered the net photosynthetic rate, transpiration rate, and stomatal conductance of plants [17,18]. These aforementioned results provide useful evidence regarding the inhibition of plants due to leaked $CO_2$.

Some work has also been undertaken to explore the below-ground root system response to $CO_2$ leakage from CCS. It has been shown that an increased soil $CO_2$ concentration inhibits root respiration and decreases root density [19,20] and that the root lengths of plants, such as clover, maize, alfalfa, and teosinte, were significantly affected by $CO_2$ leakage [21]. The total number of roots of oilseed rape decreased at depths below 20 cm in soil with high $CO_2$ levels compared to those grown in soil with no elevation of $CO_2$ [22]. Moreover, the root responses to stress have being revealed in depth. Zhou et al. concluded that drought significantly decreased the root length and root density while increasing the root diameter, as evidenced by a meta-analysis of 128 published studies [23]. Mild water stress can decrease the root length and increase the root diameter of maize [24]. Root systems play a key role in the uptake, transportation, and accumulation of water and nutrients in farmland ecosystems [25]. Despite the fact that roots are the first to be affected by $CO_2$ leakage, there is minimal understanding of plant root morphology under such conditions.

In this paper, pot experiments simulating $CO_2$ leakage with three treatments were conducted to study the response of root systems to elevated soil $CO_2$. The specific objective was to reveal how plant root systems respond to $CO_2$ leakage from CCS.

## 2. Methods and Materials

### 2.1. Location and Experimental Timeline

The experiment was conducted at a research station (40°5′41.87″ N, 116°55′26.76″ E) of the Institute of Environment and Sustainable Development in Agriculture, The Chinese Academy of Agricultural Sciences, based in Shunyi, Beijing, from June 19 to September 20, 2017 (93 days). The region has a temperate, semi-humid monsoon climate. The annual mean temperature is 11.5 °C and there are 2750 h of sunshine annually. The annual mean precipitation is 625 mm with a frost-free period of approximately 195 days. Maize hybrid (Zhongnuo No 2) was seeded on June 19 and seedings were singled out on July 3, leaving one seedling per pot. To simulate a constant leak, $CO_2$ was continuously injected from July 12 to September 20, 2017. Topsoil (classified as cinnamon type) was packed into pots and compacted to a thickness of 50 cm with an initial bulk density of 1.25 g cm$^{-3}$. The soil texture was medium loam with a pH of 8.38 and an organic matter content of 15.48 g kg$^{-1}$. The total nitrogen, total phosphorus, and total potassium concentrations of the soil were 0.37 g kg$^{-1}$, 0.61 g kg$^{-1}$, and 20.42 g kg$^{-1}$, respectively. The experiment was carried out in an area covered with a transparent roof to avoid uncontrolled water seepage due to precipitation. The plants were irrigated when the soil volume water content was < 25%. The amount of irrigation was determined by the water consumption during maize growing stages. Fertilizer management methods were carried out according to local practices and are listed in Table 1.

**Table 1.** Summary of water and fertilizer treatments in the pot experiment.

| Date | Measure | Quantity/pot |
|---|---|---|
| 19 June 2017 | Base fertilizer | N 1.5 g/$P_2O_5$ 1.5 g/$K_2O$ 1.5 g |
| 19 June 2017 | Irrigation | 15 L |
| 5 July 2017 | Irrigation | 5 L |
| 12 July 2017 | Irrigation | 5 L |
| 19 July 2017 | Irrigation | 5 L |
| 28 July 2017 | Irrigation | 5 L |
| 2 August 2017 | Irrigation | 5 L |
| 14 August 2017 | Irrigation | 5 L |
| 21 August 2017 | Irrigation | 5 L |
| 25 August 2017 | Irrigation | 5 L |
| 28 August 2017 | Irrigation | 5 L |
| 30 August 2017 | Irrigation | 5 L |
| 4 September 2017 | Irrigation | 5 L |
| 8 September 2017 | Irrigation | 5 L |
| 10 September 2017 | Irrigation | 5 L |
| 16 September 2017 | Irrigation | 5 L |

## 2.2. Pot Experiment of Elevated Soil $CO_2$ Treatments

The experiment was carried out on a manually manufactured $CO_2$ leakage platform consisting of a controlled $CO_2$ release device, $CO_2$ monitoring system, and artificial water and fertilizer management (Figure 1). The device comprised pots (soil chamber, permeable separator, and $CO_2$ chamber), a drain valve, gas meter, $CO_2$ gas source, gas duct, ball valve, and a gas shunt (Figure 1), which was manually controlled to release $CO_2$. The cultivation pot was $50 \times 50 \times 80$ cm$^3$ in size. The soil chamber (60 cm in height) and gas chamber (20 cm in height) were separated by a layer of shim with a 0.5 cm aperture, to ensure homogeneous $CO_2$ injection. The $CO_2$ gas in the bottle was injected into the gas chamber of the cultivation containers through the gas duct. Then, $CO_2$ entering the soil chamber passed through a permeable separator and reached near the root system.

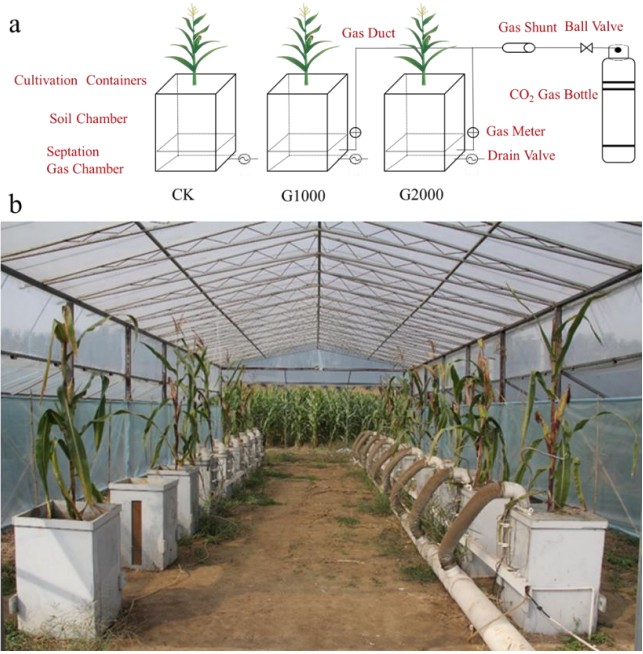

**Figure 1.** Schematic (**a**) and photograph (**b**) of the simulated $CO_2$ leakage platform in the experiment. $CO_2$ was injected into the pots from below at different controlled rates using a gas shunt and a gas meter. The $CO_2$ injection flux was set to three different values (0 g m$^{-2}$ d$^{-1}$, 1000 g m$^{-2}$ d$^{-1}$, and 2000 g m$^{-2}$ d$^{-1}$).

### 2.3. CO$_2$ Injection Treatments

The experiment included one control treatment (no added CO$_2$, CK) and two elevated soil CO$_2$ treatments (G1000, G2000). The CO$_2$ injection flux was used to define the treatments [15] as follows: CK (0 g m$^{-2}$ d$^{-1}$), G1000 (1000 g m$^{-2}$ d$^{-1}$), and G2000 (2000 g m$^{-2}$ d$^{-1}$). Each treatment had three replicates. The CO$_2$ injection rates, which were controlled by the valve opening size, on the treatment condition: CK (0 mL min$^{-1}$), G1000 (88 mL min$^{-1}$), and G2000 (176 mL min$^{-1}$). These rates were based on:

$$F = \frac{\rho v}{s} \tag{1}$$

where F is the CO$_2$ injection flux (g m$^{-2}$ d$^{-1}$), $v$ represents the CO$_2$ injection rate (mL min$^{-1}$), $\rho$ is the atmospheric carbon dioxide density under normal pressure (approximately 1.977 g L$^{-1}$), and $s$ is the cross-sectional area of the pot (0.25 m$^2$).

### 2.4. Root Sampling Preparation and Separation

The cleaned roots are shown in Figure 2a (three samples per treatment) and each nodal root was separated. The total number of nodal roots for CK, G1000, and G2000 were 78, 102, and 90, respectively. The nodal roots of the root system of maize could be divided into four nodal-root whorls depending on their initial positions and sequences of growth [26]. The nodal roots whorled from the first to the fourth were defined enough to be distinguishable, and a small number of indeterminate nodal roots were put in the fourth nodal-root whorls to analyze the whole system. Nodal roots were grouped according to the nodal-root whorls in preparation for scanning.

(a) (b)

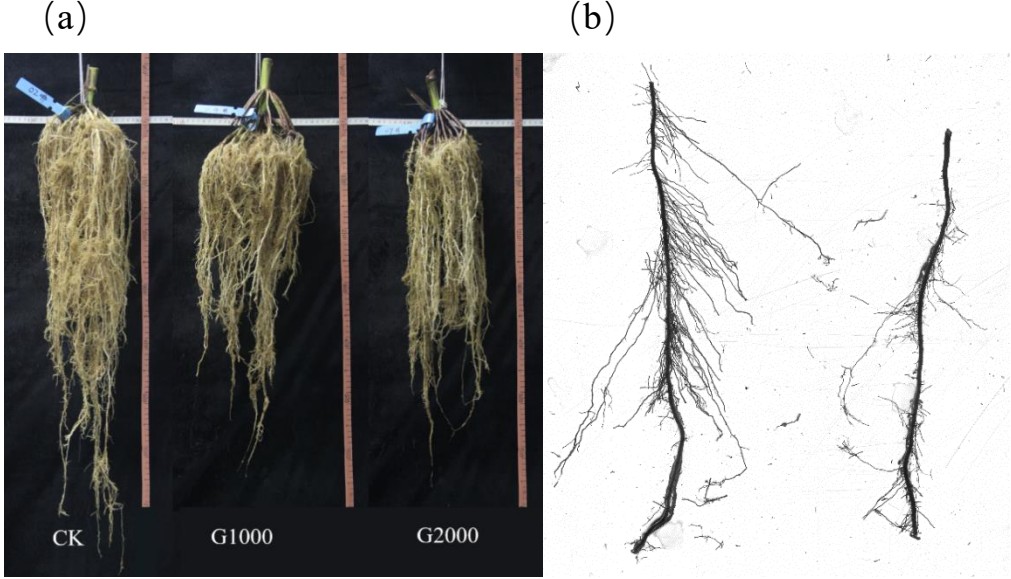

**Figure 2.** Maize root systems (**a**) from the CK and elevated soil CO$_2$ treatment and (**b**) an example of the scanned picture of the root disk with two root segments.

### 2.5. Root Morphology Scanning and Parameter Analysis

Prepared nodal roots needed to be separated into 10-15 cm long root segments before they could be placed into the root disk (Figure 2b). The number of small root segments placed in the root disk (i.e., a 200 mm × 150 mm transparent rectangular dish) was determined by the complexity of the axial branching sections. The root disk was laid on the scanner-imaging surface and filled with 1-2 mm of water. Prior to scanning, the lateral roots of each root segment were spread out in the water using tweezers to reduce root overlap. The roots were then scanned using a flatbed scanner (Epson V800) with a resolution of 400 d.p.i. Subsequently, the scanned root images were analyzed by WinRhizo 2013

Pro image-analysis software and the erasing tool was used to remove noise in the original images. Once the scanned images had been analyzed, root parameter data including length, diameter, volume, and surface area were transferred to a text file.

### 2.6. Data Analysis

All data analyses were performed using SPSS ver. 25. The differences in measured variable means of the roots between CK and the two $CO_2$ treatments were analyzed using independent-sample *t*-tests. One-way analysis of variance (ANOVA) was used to examine the difference among the root diameters of different nodal-root whorls in CK and elevated soil $CO_2$ treatments. Differences among treatments with $p < 0.05$ were considered significant.

## 3. Results

### 3.1. Root Characteristics in Response to Elevated Soil $CO_2$ Treatments

The root surface area and root length for the G2000 $CO_2$ treatment were the root morphological characteristics that were significantly affected ($p < 0.05$) by $CO_2$. There were obvious distinctions in root parameters between each treatment (Table 2); the values of the root length, root surface area, and root volume for G2000 were lower than that of CK, while the values of the root volume for G1000 and the values of the root diameter for the $CO_2$ treatments were not less than CK treatments.

**Table 2.** The values of root length, volume, surface area, diameter, and dry weight in CK and two elevated soil $CO_2$ treatments. *t*-test results for root parameters (mean ± SE) and their percentage decreases in CK and leakage treatments. ※: The significant $p$ values, marked with asterisks, indicate significant differences between CK and elevated soil $CO_2$ treatments ($p < 0.05$). The $p$ values between two $CO_2$ treatments of all root parameter are all greater than 0.5.

| Root Parameters | Mean Values CK | Mean Values Treatments | $t$ | $n$ | $p$ | Decrease (%) |
|---|---|---|---|---|---|---|
| Root length (cm) | 37,924.62 ± 3777.91 | 35,047.00 ± 7606.10 (G1000) | 0.34 | 3 | 0.75 | — |
| | | 20,960.99 ± 2860.17 (G2000) | 3.58 | 3 | 0.02※ | 44.73% |
| Root volume (cm$^3$) | 65.34 ± 0.53 | 71.99 ± 12.06 (G1000) | −0.51 | 3 | 0.61 | — |
| | | 52.82 ± 4.26 (G2000) | 2.92 | 3 | 0.43 | — |
| Root surface area (cm$^2$) | 5444.83 ± 266.16 | 5420.61 ± 983.64 (G1000) | 0.24 | 3 | 0.98 | — |
| | | 3585.8735 ± 368.70 (G2000) | 4.09 | 3 | 0.02※ | 34.14% |
| Root diameter (mm) | 0.047 ± 0.0022 | 0.051 ± 0.0016 (G1000) | −1.6 | 3 | 0.19 | — |
| | | 0.057 ± 0.0030 (G2000) | −2.65 | 3 | 0.06 | — |
| Root dry weight (g) | 19.37 ± 2.0 | 24.65 ± 3.3 (G1000) | −1.37 | 3 | 0.246 | — |
| | | 18.73 ± 1.34 (G2000) | 0.28 | 3 | 0.80 | — |

The mean root length and root surface area for the G2000 decreased by 34.14% and 44.73%, respectively, in comparison with the CK treatment (Table 2, $p < 0.05$). However, the mean values of the root diameter for the $CO_2$ treatments G1000 and G2000 increased by 8.51% and 21.28%, respectively, compared with the CK treatment. The results indicate that the mean root length was shorter when root systems were chronically exposed to excess $CO_2$. In addition, the root diameter increased only slightly in the G1000 and G2000 treatments compared with the CK treatment.

### 3.2. Development of the Root Diameter Under Different Levels of Elevated Soil $CO_2$

A total of nine sampled root systems from the CK, G1000, and G2000 treatments were scanned. These images were of whole root systems; the numbers of scanned pictures for the $CO_2$ treatments G1000 and G2000 were 179 and 111, respectively, representing decreases of 3.7% and 40.3%, respectively, compared with the CK treatment (186). The nodal roots of the maize root system could be divided

into four nodal-root whorls depending on their initial positions and sequences of growth in the CK and $CO_2$ treatments (Figure 3). The median basal diameter of the first, second, and fourth nodal-root whorls in the CK were significantly smaller than those in the $CO_2$ treatments (Figure 3, $p < 0.05$), while the differences in median basal diameter between the second nodal-root whorl lateral roots (LRs) were not significant (Figure 3, $p > 0.05$).

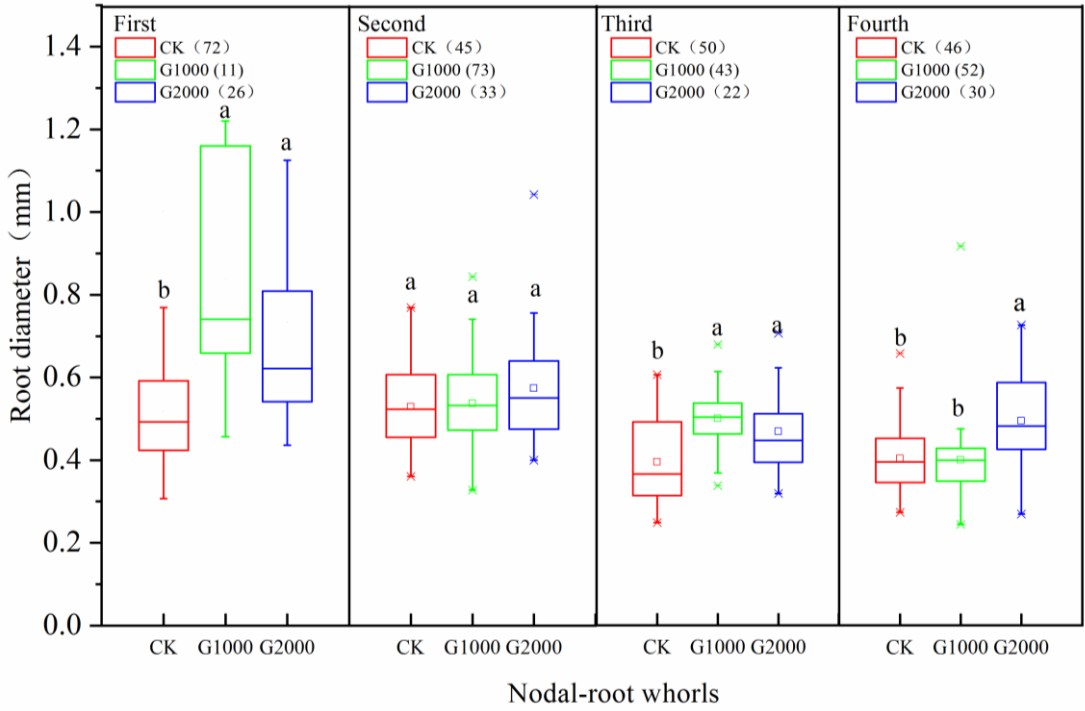

**Figure 3.** Basal diameter variation of different nodal-root whorls of maize in the control (CK) and elevated soil $CO_2$ treatments (G1000 and G2000). Medians (solid, lines in box), and 25% and 75% quartiles (lower and upper margins of the boxes) of nodal-root whorl diameters are shown. Different letters (e.g. a, b) indicate significantly difference at a 5% significance level. The total number of root disks of each root system is given in the key.

### 3.3. Relative Diameter Class Length Under Different Levels of Elevated Soil $CO_2$

The relationships between root length and basal diameters of the whole root systems for CK and the $CO_2$ treatments were determined. As shown in Figure 4a, the data was fitted by an exponential equation, and in Figure 4b, the data, of which root length could be log-transformed, was also fitted with a linear equation.

$$Y = a + bX \tag{2}$$

Here, Y refers to the root length after being log-transformed, and X refers to diameter grouped from 0 to 2.9 mm at intervals of 0.1 mm. The mean values of root diameters are represented by X (x1, x2, . . . , x30), and the histograms refer to the mean values of the root length Y (y1, y2, . . . , y30). The values of $a$ in the exponential equation for CK, G1000, and G2000 were 3.651, 3.432, and 3.351, respectively; in this regard, G1000 and G2000 were less in comparison with the CK treatment. The values for $b$ were −0.824, −0.74, and −0.744, for CK, G1000, and G2000, respectively; in this regard, G1000 and G2000 were greater compared with CK treatments. The correlations were significant, and the coefficients of determination ($R^2$) were all greater than 0.80 ($R^2 = 0.924$ for CK, $R^2 = 0.871$ for G1000, and $R^2 = 0.890$ for G2000). The slope of the exponential equation for the CK treatment was steeper than that for the $CO_2$ treatments, and the values of $b$ of linear equations in $CO_2$ treatments were greater compared with the CK treatment, which also demonstrated that the increased $CO_2$ flux was consistent with a great decrease in the root length of the small-diameter class (Figure 4b).

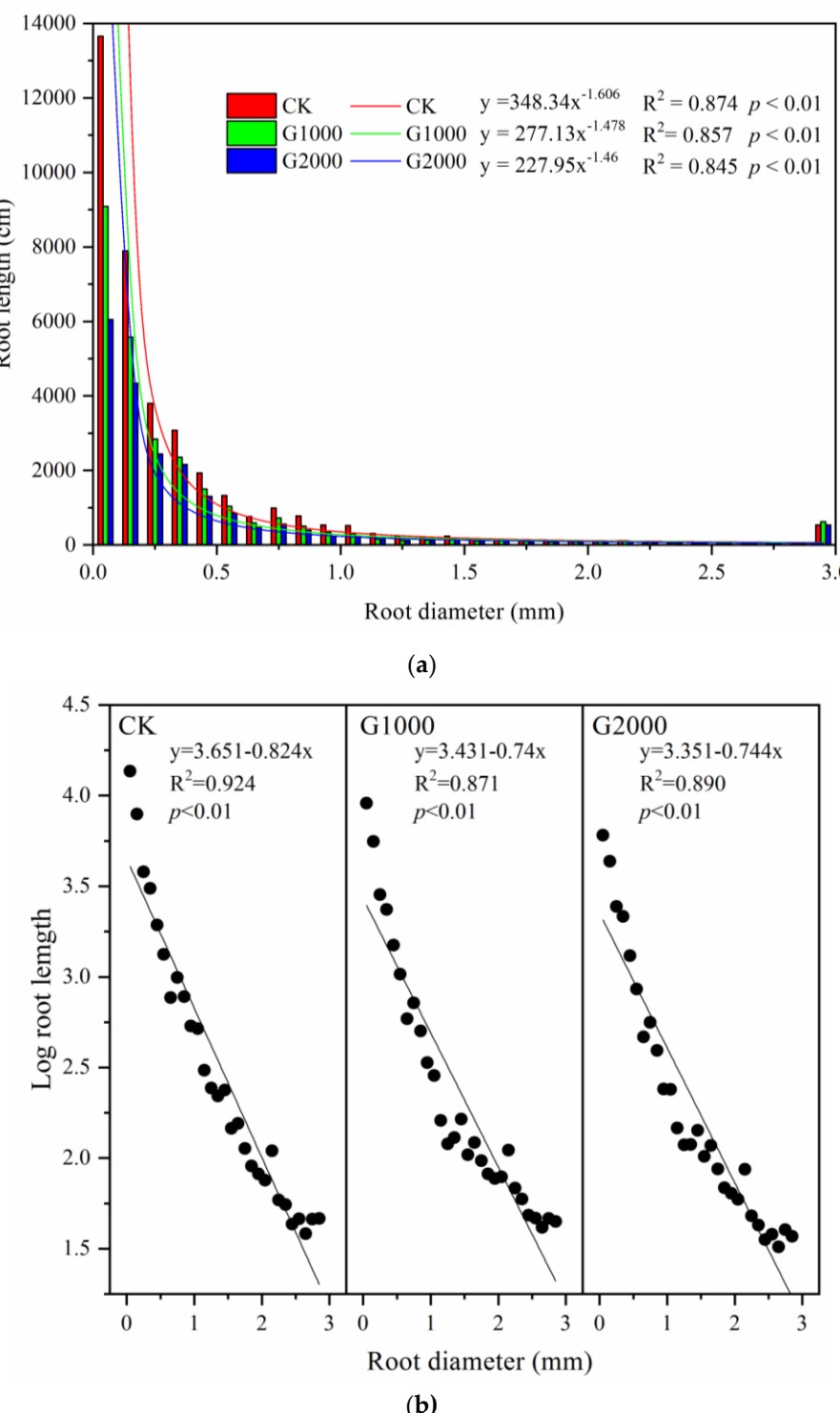

**Figure 4.** Histogram of the maize relative diameter class length by diameter class in control (CK) and elevated soil $CO_2$ treatments (G1000 and G2000). (**a**) Each bar represents the total root length of the corresponding diameter (**a**). Basal diameters were divided into 29 classes, from 0.1 to 2.9 mm at intervals of 0.1 mm. The value of the diameter in the 29th class was over 2.9 mm. These images were of the whole root systems. (**b**) The data was fitted with a linear equation for which root length was to be log transformed.

For root lengths in the first 20 diameter classes (i.e., 0.1–2.0 mm), those of the $CO_2$ treatments were less than in the CK treatment (Table 3, $p < 0.05$), while the root length did not show a significant

difference in the large-diameter class (d > 2.0 mm) in comparison with the CK treatment (Table 3, $p > 0.05$). The average fine root lengths for G1000 and G2000 within the root diameter range 0–1 mm were 24,529.3 ± 1325.5 cm and 18814.3 ± 2735.3 cm, respectively, representing decreases of 29.44% and 45.88%, respectively, in comparison to the CK treatment (Table 3, $p < 0.05$). The mean values of root lengths within the root diameter range of 1–2 mm for G1000 and G2000 were 1341 ± 85.7 cm and 1202.8 ± 83.9 cm, respectively, representing decreases of 36.37% and 42.92%, respectively, compared to the CK treatment (Table 3, $p < 0.05$). The mean root lengths of G1000 and G2000 for diameters of 2–2.9 mm were not significantly different from those of the CK treatment (Table 3).

**Table 3.** Impact of different $CO_2$ treatments on root length (mean ± SE) of different diameter classes. d represents root diameter (mm). The numbers with the same letters are not significantly different at a 5% significance level.

| Treatments | Root Length (cm) | | | |
| --- | --- | --- | --- | --- |
| | 0 < d ≤ 1.00 | 1.0 < d ≤ 2.00 | 2.0 < d ≤ 2.90 | d > 2.90 |
| CK | 34763.6 ± 3769.6a | 2107.4 ± 74.0a | 519.4 ± 34.3ab | 532.6 ± 41.3a |
| G1000 | 24529.3 ± 1325.5b | 1341 ± 85.7b | 544.8 ± 22.2a | 625.5 ± 58.5a |
| G2000 | 18814.3 ± 2735.3b | 1202.8 ± 83.9b | 419.3 ± 34.8b | 533.0 ± 47.3a |

## 4. Discussion

This study examined the morphology of maize root systems exposed to simulated $CO_2$ leakage from CCS. The significant decrease of the total root length induced by elevated soil $CO_2$ indicated that leaked $CO_2$ could inhibit root growth. More specifically, the shortening of the total root length was mainly because of the reduction of fine root length (Table 2). Kim et al. demonstrated that root length was significantly less (by 47%) in $CO_2$ gas injection treatments (with an introduced flow rate of 250 mL min$^{-1}$) relative to the control [27]. We confirmed that the mean root lengths for the elevated soil $CO_2$ (G1000 and G2000) treatments were reduced by 7.59% and 44.73%, respectively, compared to the CK treatment. Our findings suggest that the knowledge of the decrease in fine root length is essential for explaining how plants respond to $CO_2$ leakage from CCS.

Apart from the discussion of the morphological characteristics of the total root system for the CK and elevated soil $CO_2$ treatments, it is important to highlight the root diameter, as this would demonstrate a noteworthy and interesting response to leaked $CO_2$ in soil. The median basal diameter of nodal-root whorls was substantially lower in CK than those in the elevated soil $CO_2$ treatments (Figure 3, $p < 0.05$). The diameter results for each nodal-root whorl was equivalent to the mean diameter changes of total root systems (Figure 3), indicating that there were fewer lateral roots in the $CO_2$ treatments than in CK. Additionally, root diameter and length distribution are important features of root systems [28]. Many studies have examined how the diameter class lengths of plants are affected by different fertilization measures [29–31], however, more information is needed to understand the effects of elevated $CO_2$ levels on plant roots. Figure 4 shows the exponential relationship between root length and basal diameter classes, and the observed significant root length and diameter change between CK and $CO_2$ treatments. Injected $CO_2$ resulted in reduced root lengths and increased root diameters. Other studies of maize roots have shown that $CO_2$ leakage produced adventitious roots [20], the prop roots of maize arising from the lower stem, which would increase the mean root diameter. However, the model in our study was different from that of Zobel et al. who used the non-linear regression extreme value model to fit the diameter class length (DCL; mm cm$^{-3}$) [29]. Possibly due to different scanner resolutions, the minimum class of the maize root diameter was 0.1 mm in our study and 0.01 mm in the Zobel et al. study. Furthermore, it should be noted that the present study scanned all the roots of maize without distinguishing between above-ground nodal roots (adventitious roots) and below-ground nodal roots. This may be a possible explanation for the fact that the root lengths with a diameter >2.9 mm did not show a consistent trend regarding the root diameter-class distribution (Figure 4a, Table 3).

A more detailed analysis of the root length under different diameter classes was carried out through the comparison of measurements in CK and $CO_2$ treatments (Table 3). In this study, root size fractions were based on previous investigations, where a diameter of 1 mm was used to distinguish coarse and fine root fractions [30]. In fact, it is widely recognized that environmental factors determine the production and turnover of roots with a diameter < 2 mm [32]. As shown in Table 3, the proportion of root lengths with diameters < 2 mm were all > 95%. It is generally considered that fine roots represent 90% or more of the total root length of a given mature plant [29]. Additionally, root length with a smaller diameter was significantly affected by elevated $CO_2$ levels in soil, whereas coarse roots with a diameter > 2 mm were not consistently influenced. Therefore, it could be concluded that the fine roots were the most sensitive indicator of leaked $CO_2$.

## 5. Conclusion

This study focused on the response of maize root systems to simulated $CO_2$ leakage. The significant decrease in the total root length induced by elevated soil $CO_2$ indicated that leaked $CO_2$ could inhibit root growth. The total root length was highly sensitive to $CO_2$ when compared to other root characteristics such as root volume, root diameter, root surface, and dry root weight. More specifically, the shortening of the total root length was mainly because of the reduction of the lengths of fine roots. Our findings suggest that understanding the decrease in fine root length is essential for explaining how plants respond to $CO_2$ leakage from CCS.

**Author Contributions:** Conceptualization and methodology, X.Z.; writing—original draft preparation, Y.H.; writing—review and editing, X.M. All authors have read and agreed to the published version of the manuscript.

**Funding:** This research was supported by National Key R&D Program of China (No. 2018YFC1508805 and 2016YFC0500508), the National Natural Science Foundation of China (No. 31600351), and the Strategic Priority Research Program of Chinese Academy of Sciences (No. XDA 20010302). We are grateful to Xiang Ji, Mengying Yu, and Yuzheng Lu for their important contributions in field management and measurements.

**Conflicts of Interest:** The authors declare no conflict of interest.

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
