# Peer review of "Fine Root Length of Maize Decreases in Response to Elevated CO2 Levels in Soil"

_applsci, doi:10.3390/app10030968_

Round 1
Reviewer 1 Report
The terminology that underpins the description of the study is not quite correct. The authors suggest that they have a control; no CO2 gas. Strictly speaking this is not an experimental control. In the experimental system described it is not possible to separate out a CO2 signature from an O2depletion signal. This issue of expertemental design has been discussed in Lake et al (2017) International Journal of Greenhouse Gas Control 64, 333-339. The units used to express the leakage are also fairly odd making it very difficult for the reader to compare the present study to the published literature. It would make more sense to present the treatments as a flux rate which is shown later in the paper and to report the CO2 concentration in the results as percentage CO2. This will make it much easier for the reader to contextualise the data and for the authors to ground their data within the available literature.
In the abstract the authors comment that relatively few studies have focused on roots. I’m not sure I agree with this statement and I don’t think it reflects fully the nature of the work published on CCS/plant research. The paragraph on page 3 lines 54—78 is very poorly referenced. Numerous plant response are mentioned but these inferences are not supported by any references. This short fall needs to be addressed in any resubmission. The statement about a lack of below ground data (lines 73—78) doesn’t fully align with the state of the published literature. A number of workers from various CCS field sites have looked at below ground activity for example the work at ASGARD (Artificial Soil Gassing And Recovery Detection) at the University of Nottingham has done much to explore below surface plant responses and this should be acknowledged.
In the methods there is no mention of pot size; are the authors confident that the control plants did not become pot bound with root inhibition possibly influencing their data? The authors have presented a number of different root measurements but the rationale behind the selection of these parameters is missing. Consequently it is difficult to establish why these data were collected. It would be helpful if a rationale could be introduced to frame the data collection. The discussion and analysis fig 4 is unclear.
Did the authors see a stress response in the above ground plant tissue that reflects the loss of fine roots
Reviewer 2 Report
Statistical approach and reporting language have some serious problems, but the experimental approach was good and the results meritworthy. Please refer to the attached file for details.

Reviewer 3 Report
The reviewer thanks the authors for considering this journal for publication of their research.
The manuscript reports on the results of a greenhouse/controlled enviroment study of the impact of CO2 treatments on maize root growth and morphology. The reported results are interesting and do add to the published literature. I have suggestions on how to improve the manuscript; these should be addressed prior to publication. I have divided my comments into major and minor.
Major Comments:
The description of the experiment in Methods is inadequate. The authors should include information about the specific maize variety and the greenhouse conditions (temp, lighting, humidity). Information should also be provided about the watering protocol, e.g., what decision process was used to determine if to water? In the results section is unclear if the appropriate statistical tests are being used. What is the null hypothesis? If there are three treatments - control, G1000, and G2000 - would it make more sense to do ANOVA across treatments and follow up with posthoc testing to determine which treatments are significantly different? The way the results are presented in Table 2 gives the impression that the null hypothesis is one sided. Is this the case? In the results section there are many places where the authors claim a decrease or increase, with no statistical results. See, for example, both sections labeled Section 3.2 It should be very clear when a statement is based on statistical significance and when a statement is simply a description of the results. The exponential function modeling in section 3.2 and figure 4 is poorly explained. Usually the data would be log transformed and then modeled with linear regression. Also, Figure 4 is not a histogram as the y axis is not frequency. I'm not sure what is being shown. Does each bar reflect one observation or a mean. The authors claim that `correlations were significant' and `root length ... of the leakage treatments ... all decreased' without any statistics. All claims should be supported by statistics. Figure 3 should include the results of statistical testing between the groups. In table 3 it is unclear what tests the different letters represent. I assume that, for example, CK root length 2<d<=2.90 is not significantly different from any other group?
Minor Comments:
The manuscript would benefit from a read through by a native speaker of English. The statement prior to reference 5 should be supported by additional references. The reference on line 128 should be shown as (10) The last sentence (lines 176-178) should be removed. How do the number of scanned pictures (see line 196) relate to root characteristics?
